

# Effect of predation risk and ectoparasitic louse flies on physiological stress condition of the red-tailed tropicbird (*Phaethon rubricauda*) from Rapa Nui and Salas & Gómez islands

Nicolas Luna[1,2], Andrea I. Varela[1], Guillermo Luna-Jorquera[1,3] and Katherina Brokordt[3,4]

[1] Millennium Nucleus for Ecology and Sustainable Management of Oceanic Islands (ESMOI), Departamento de Biologia Marina, Universidad Católica del Norte, Coquimbo, Chile
[2] Programa de Magister en Ciencias del Mar Mención Recursos Costeros, Facultad de Ciencias del Mar, Universidad Católica del Norte, Coquimbo, Chile
[3] Centro de Estudios Avanzados en Zonas Áridas (CEAZA), Universidad Católica del Norte, Coquimbo, Chile
[4] Laboratorio de Fisiología y Genética Marina (FIGEMA), Departamento de Acuicultura, Universidad Católica del Norte, Coquimbo, Chile

Corresponding author
Andrea I. Varela, and.vrl@gmail.com

## ABSTRACT

**Background:** Introduced predators at seabird colonies and parasites may have lethal and/or sub-lethal consequences for bird populations. We assessed the potential sub-lethal negative effects of these stressors in a native seabird listed as vulnerable in its south-eastern pacific distribution. This study was conducted in two red-tailed tropicbird (*Phaethon rubricauda*) colonies, one located in Rapa Nui Island, which is threatened by the presence of introduced predators, and the other located in Salas & Gómez Island, which has no introduced predators, but birds are infested by ectoparasitic louse flies.

**Methods:** The effects on physiological stress traits of both, predation risk on different nest types (protected and exposed) on Rapa Nui, and different levels of louse flies' parasitic loads on Salas & Gómez were studied. Three variables were analyzed: the heterophil/lymphocyte (H/L) ratio, the transcriptional levels of mRNA *HSP70* in blood, and the body condition. These stress indicators and leukocyte counts were compared between colonies.

**Results:** No significant differences were found in any stress indicator between different nest types within Rapa Nui, showing that the effect of predator's presence was the same for adults nesting in both, protected and exposed nests. No significant correlation was found between louse flies' parasitic loads and any stress indicators in the birds of Salas & Gómez. Also, there was no difference in any stress indicator between islands. However, a significant opposite trend between islands was found in the eosinophil, lymphocyte, and monocyte counts when related to body condition.

**Conclusions:** We found a lack of significant differentiation in all the stress level indicators assessed within and between islands. The presence of louse flies in Salas & Gómez vs. the absence of this parasite at Rapa Nui may be the cause for the significant difference in the trend of eosinophil, lymphocyte and monocyte counts between the islands. However, further studies are necessary to elucidate the reason

for this difference and to better investigate the lethal effects of introduced predators on the Rapa Nui colony to evaluate appropriate conservation measures for this native seabird.

## INTRODUCTION

Seabirds on most oceanic islands have evolved in the absence of terrestrial predators (*Coulson, 2002*), and predation can be due to other seabirds (*Baird, 1996*), and secondarily by raptors birds like owls and hawks (*Glue, 1972*; *Hamer, Schreiber & Burger, 2002*). There is good evidence that alien or introduced terrestrial predators are a severe threat to a wide variety of seabird species (*Coulson, 2002*; *Caut, Angulo & Courchamp, 2008*; *Jones et al., 2008*). The most typical introduced predators are rats, especially from the genus *Rattus*, feral cats and dogs, among others (*Nogales et al., 2004*; *Krajick, 2005*; *Caut, Angulo & Courchamp, 2008*; *Jones et al., 2008*; *Croxall et al., 2012*). Predators may affect seabird populations by lethal interactions (e.g., predation) and/or non-lethal interactions as the result of predation threat (*Cresswell, 2008*). Non-lethal interactions can alter birds at the ecological level. For example, foraging seabirds can avoid suitable feeding areas in response to predation risk (reviewed by *Cresswell, 2008*), but also may have other negative physiological consequences. For example, predator vocalizations were interpreted as a danger cue by the Neotropical blue-black grassquits (*Volatinia jacarina*), inducing dramatic changes in the proportion of heterophils and lymphocytes, and generating immune depression (*Caetano et al., 2014*). Similarly, the presence of predators increased stress protein expression in late breeding pied flycatchers (*Ficedula hypoleuca*) (*Thomson et al., 2010*). These studies support the idea that predation risk may affect the physiological condition of prey organisms at different levels. So far, to the best of our knowledge, studies regarding the potential negative effects of predation risk on the physiological condition of seabirds are lacking.

Like predation, parasitism has negative lethal and sub-lethal effects on birds (*Boyd, 1951*). Birds are exposed to a wide variety of ectoparasites, including biting lice (Mallophaga) fleas (Siphonaptera), bugs and triatomines (Hemiptera) and flies (Diptera) (*Boyd, 1951*). Parasites can decrease physiological status of their host birds by altering their immune system, changing, for example, white blood cell proportions (*Davis, Maney & Maerz, 2008*), and stress protein levels in blood (*Martínez-de la Puente et al., 2011*). They may have further deleterious effects on their host's life history traits, affecting growth (*Fitze, Tschirren & Richner, 2004*), breeding success (*Møller, 1993*; *Martínez-de la Puente et al., 2011*), and survival of birds (*Boyd, 1951*), and can even drive hosts to local extinction (*Koop et al., 2016*). In particular, flies from the family Hippoboscidae (order Diptera), commonly called "louse flies", are obligate ectoparasitic blood feeders (*Rahola, Goodman & Robert, 2011*) and have been reported to parasitize at least 18 different orders

of birds (*Santos, López & Miller, 2014*). Species of the Hippoboscidae are also known to act as infectious vectors in transmitting haemosporidian blood parasites to seabirds (*Rahola, Goodman & Robert, 2011*), thus activating the host immune system (*Padilla et al., 2006*; *Levin et al., 2011*). Further, high parasitic loads by louse flies have been shown to decrease avian fitness by reducing the breeding success of adults and reducing the weight, size and survivorship in nestlings (*Loye & Carroll, 1995*).

A widely used methodology to assess stress levels in birds, and many taxa among vertebrates, is the examination of leukocyte profiles from blood smears (*Davis, Maney & Maerz, 2008*). Some leukocytes are altered by stress hormone levels, and thus are indirectly associated with stress status (*Davis, Maney & Maerz, 2008*). Among leukocyte proportions, the estimation of the ratio between circulating heterophils and lymphocytes (H/L ratio) is commonly used as a stress indicator, because heterophils (H) and lymphocytes (L) are altered by stress in opposite directions, and so the H/L ratio varies with stress levels (*Maxwell & Robertson, 1998*; *Quillfeldt et al., 2008*; reviewed by *Davis, Maney & Maerz (2008)*). Further, the relative number of monocytes and eosinophils are also important to discriminate whether H/L ratios are high due to stress, disease or parasites. Monocytes due to their potential phagocytic function increase their proportion due to infections or intracellular parasites such as viruses and certain bacteria, while eosinophils increase due to infections with parasites such as worms and protozoa (*Maxwell, 1993*; *Davis, Maney & Maerz, 2008*).

Another indicator of physiological stress in birds is the expression levels of the heat shock proteins (HSPs). Expression levels of HSPs are induced by different factors, like heat/cold shock, physical activity, and toxins (*Sørensen, Kristensen & Loeschcke, 2003*; *Brun et al., 2008*). Further, changes in these proteins in the blood have been related to bird responses to stress caused by low food availability, social interactions, predation risk and parasites, among other factors (*Tomás et al., 2004*; *Thomson et al., 2010*; *Martínez-de la Puente et al., 2011*). A positive correlation between *HSP60* levels in blood and ectoparasite infestation was observed in a manipulative experiment with nesting blue tits *Cyanistes caeruleus* (*Martínez-de la Puente et al., 2011*). Similarly, an increase of *HSP70* expression levels was found in nesting pied flycatchers (*Ficedula hypoleuca*) due to increasing predation risk (*Thomson et al., 2010*).

Body condition is usually a reflection of the physiological status and is commonly used as a proxy for an individual's energy stores (*Jacobs et al., 2012*). It can be estimated as a ratio between observed body mass and an expected body mass calculated for each bird using morphological measurements (*Plischke et al., 2010*). Researchers mostly relate condition index to breeding success, bird survival, and behaviour (*Brown, 1996*; *Labocha & Hayes, 2012*), and it is generally accepted that a "good body condition" is synonymous of a "healthy animal". Therefore, there are many attempts especially in physiological research to test the hypothesis that adverse environmental conditions negatively affect the body condition (e.g., *Thomson et al., 2010*), as responding to these conditions involves energy-consuming processes (*Suorsa et al., 2004*; *Cresswell, 2008*; *Dehnhard, Quillfeldt & Hennicke, 2011*; *Labocha & Hayes, 2012*).
A model species to assess the potential pervasive effects of predation risk and parasitism in seabirds that breed at oceanic islands is the red-tailed tropicbird, *Phaethon rubricauda*. This tropicbird is one of the three species of the Family Phaethontidae (Order Phaethontiformes); it occurs in tropical and sub-tropical areas of the Pacific and Indian Oceans and breeds on oceanic islands and islets (*Del Hoyo, Elliott & Sargatal, 1992*). They are monogamous, ground-nesters, and both sexes share reproductive duties evenly (*Fleet, 1974*). In Chile, in the south-eastern Pacific Ocean, the species breeds only in Rapa Nui (also known as Eastern Island), Salas & Gómez and Desventuradas Islands (*Aguirre et al., 2009*; *Flores, Schlatter & Hucke Gaete, 2014*), where it has been categorized as a vulnerable species (*Ministry of the Environment, Chile, 2017*).

The present study was conducted in two red-tailed tropicbird colonies, one located in Rapa Nui and the other in Salas & Gómez. Rapa Nui is an inhabited island located ~3,700 km from mainland Chile (27°09′S, 109°26′W). The assessed colony is located at the Rano Raraku volcano in the eastern part of the island. This colony is almost free of louse flies (N. Luna, 2016, personal observation), but introduced predators are present, including rats of the genus *Rattus*, feral cats, feral dogs and the raptor bird chimango caracara (*Phalcoboenus chimango*) (*Flores et al., 2017*; *Luna et al., 2018*). Two distinctive red-tailed tropicbird nest types were recognized in this colony, one covered by rocks (sometimes moais or inside caves) characterized as "protected", and the other type covered only by grass, with little or no rocky protection, characterized as "exposed" (as describe *Prys-Jones & Peet, 1980*). *Luna et al. (2018)* registered the presence of introduced rats in 75% of the assessed nests, while avian predators only visited an exposed nest, suggesting that exposed nests might be more vulnerable, as they are also threatened by aerial predators in addition to terrestrial predators.

In contrast, Salas & Gómez Island is a small uninhabited island, located 390 km east of Rapa Nui (26°27′S, 105°28′W) within the Motu Motiro Hiva Marine Park. Unlike the Rapa Nui colony, at this island red-tailed tropicbirds nest on the ground, mostly under rocky protection, probably because of the lower presence of grass and vegetation (N. Luna, 2016, personal observation). This island has no native or introduced terrestrial predators, but louse flies have been found in most assessed red-tailed tropicbird individuals, some of them with more than 20 flies (N. Luna, 2016, personal observation).

The aim of this study was to determine whether both the presence of introduced predators in the Rapa Nui colony, and parasitism by louse flies in the birds of Salas & Gómez affect the stress levels of red-tailed tropicbirds. Three main variables were analyzed to assess the physiological stress: (a) the H/L ratio, (b) the transcriptional levels of *HSP70* in blood and (c) the body condition. Furthermore, these stress indicator levels were compared between both colonies, in order to evaluate which stressors (i.e., predation risk vs. parasitic infestation) had a higher effect on the physiological and condition status of red-tailed tropicbird populations in the Rapa Nui ecoregion. Results could contribute to better understand the sub-lethal effects of introduced predators and parasites on native species.

Three main hypotheses were evaluated in this study: (1) Red-tailed tropicbirds of the Rapa Nui colony nesting in nests with potentially higher predation risk (exposed) will show higher levels of physiological stress and lower body condition than conspecifics

nesting in nests with lower predation risk (protected); (2) Red-tailed tropicbirds of the Salas & Gómez colony with higher louse flies' parasitic load will show higher levels of physiological stress and lower body condition than conspecifics with lower parasitic load; (3) Red-tailed tropicbirds of the Salas & Gómez colony will show higher physiological stress and lower body condition due to chronic parasitic infestation than the red-tailed tropicbirds of the Rapa Nui colony where the presence of predators at nests is not permanent.

## MATERIALS AND METHODS

### Sample collection

In both colonies, nests are scattered through the breeding area and just a few are next to each other. A total of 51 red-tailed tropicbird adults were captured at their nests by hand and assessed for blood analyses and body condition. Twenty-six birds were sampled on Rapa Nui and 25 on Salas & Gómez. In Rapa Nui, nine birds were assessed in August 2016, and 17 birds were assessed in June 2017. Of these 26 birds, 20 were incubating one egg, one was rearing a recently hatched chick and five had neither egg nor chick on their nests. On June 2017, nests were categorized as "protected" (totally or partially covered by rocks or moais or inside crevices) and as "exposed" (covered by grass only). On Salas & Gómez nine birds were assessed in August 2016, nine in November 2016, and seven in June 2017. Of these 25 birds, 19 were incubating one egg, two were rearing a chick and four had neither egg nor chick on their nests.

Data of all birds assessed in both islands were used to compare physiological stress indicators and body condition between islands. Because only two birds from Rapa Nui captured in August 2016 had louse flies (each having only one fly), the evaluation of physiological stress for this colony only considered predation risk at different nest types as explanatory variable and not parasitic load. Similarly, because of the absence of introduced terrestrial or avian predators on Salas & Gómez, the parasitic load alone was considered as the explanatory variable for this colony.

Prior to blood sampling and body condition assessment, birds were examined for the presence of louse fly parasites. Flies inhabit the bird and they move over and between the feathers, therefore they can be captured by hand, searching them between the feathers. When flies flew away, they quickly returned to the host, which facilitated their capture. All flies were removed manually within the first 5 min of handling and stored in a zip lock plastic bag with acetate soaked cotton to kill them, and then preserved in 90% ethanol for later quantification and identification. Parasitic load was considered as the absolute number of flies removed from each bird within 5 min of examination, always by the same observer. Flies were identified according to *Bequaert (1941)*, *Maa (1966)*, and *Santos, López & Miller (2014)*.

### Blood sampling and body condition assessment

During manipulation, the head of the captured birds was covered with a cotton bag to reduce their vision and therefore reduce manipulation stress. A 300 µL blood sample was taken from the right wing brachial vein with a hypodermic syringe. 200 µL of blood was

distributed evenly in two cryogenic tubes containing one ml of Trizol RNA preserving reagent (Ambion®, Austin, Tx, USA), and homogenized for later determination of mRNA *HSP70* relative transcriptional levels. Samples were kept in a cool box with icepacks until being stored at 4 °C during the field trip and at −80 °C once at the laboratory. Additionally, a drop of blood was smeared on two glass slides and air-dried for later leukocyte count analysis. Smears were fixed with absolute methanol and stained with Giemsa for 30 min, following *Dehnhard, Quillfeldt & Hennicke (2011)*. After the blood sampling procedure, morphological measurements were taken for each bird. Bill, skull and tarsus length were measured using a caliper (accuracy of 0.05 mm). Wing length was measured to the nearest 1 cm with a ruler, as the distance of the longest primary from the radio-carpal joint to the tip. Morphological measurements data of all birds assessed can be found as Supplemental Material (Table S1). Each bird was then introduced in a cotton fabric bag and weighed using a 1,000 g (10 g accuracy) Pesola scale. The whole manipulation lasted 10 min maximum and birds were always returned in good conditions to their nests.

The manipulation of the birds, including blood extraction, was authorized by the Agricultural and Livestock Service (SAG), Chile, trough the permits "Resolución exenta N°9894/2015" and "Resolución exenta N°5343/2016". The Ethics committee of the Universidad Católica del Norte, Coquimbo, Chile, granted Ethical approval to carry out this study (Resolución F.M. N°12). Access to the colony at the Rapa Nui National Park was authorized by the indigenous community of Ma'u Henua and by the National Forest Corporation-Rapa Nui (CONAF-Rapa Nui).

## Leukocyte analysis and H/L ratio

Blood smears were examined under a light microscope (100×, oil immersion) in a monolayer sector of the smear. Two samples, one from Rapa Nui and one from Salas & Gómez, were excluded from leukocyte analysis as the smears were in deficient conditions.

Two smears were analyzed for each bird. Following the method described by *Davis (2005)*, white blood cells were quantified and grouped into five different leukocyte types (heterophils, lymphocytes, eosinophils, monocytes and basophils) until a minimum of 100 leukocytes per slide had been registered. Total erythrocytes were calculated as a mean of all erythrocytes counted in six microscope visual fields multiplied by the number of visual fields that were scanned to obtain 100 leukocytes. The estimation of the relative number of each leukocyte type was calculated as the percentage of all leukocytes following the method proposed by *Davis (2005)*. All five leukocyte types were standardized to 10,000 erythrocytes: each leukocyte type was multiplied by 10,000 and then divided by total erythrocytes. To calculate the H/L ratio for each bird, the mean of each standardized leukocyte of both smears was calculated, and then the H/L ratio was determined as heterophils in 10,000 erythrocytes divided by lymphocytes in 10,000 erythrocytes.

## HSP70 transcriptional level determinations

Total RNA was extracted from each blood sample using the Trizol reagent (Ambion®, Austin, TX, USA) following the manufacturer instructions. Total RNA was quantified

with an Epoch spectrophotometer (BioTek, Winooski, VT, USA) using a plate for microvolumes Take3™ (BioTek, Winooski, VT, USA) and the purity was evaluated by the A260/280 ratio (values between 1.8 and 2.0 indicate that the extracted RNA is free of DNA and phenols). Intactness was verified by visual inspection of RNA bands in denaturing formaldehyde/agarose gel electrophoresis stained with SYBR® Safe (Thermo Fisher Scientific, Waltham, MA, USA). Samples were considered intact if 28S and 18S rRNA bands were sharp and discrete with an absence of smearing under each band, and fluorescence intensity of the 28S rRNA band appeared to be twice as intense as the 18S rRNA band. After this inspection, of the total 51 samples, nine were discarded for *HSP70* transcriptional level determination. Extracted RNA was stored at −80 °C for further use. In order to obtain cDNA, reverse transcription (RT) of RNA from extracted RNA was carried out with a PrimeScript™ RT Reagent Kit with gDNA Eraser (TaKaRa, Japan) and oligo-p (dT) 15 primer, following manufacturer instructions. RT of RNAs was done in equiproportions (i.e., from equal quantity of RNA) within all compared samples from each experiment.

To isolate a partial sequence of an inducible *HSP70* for *P. rubricauda*, a pair of primers was designed using a complete sequence of *HSP70* from *Phaethon lepturus* (GeneBank accession number XM_010294882), using Primer 3 input (*Untergasser et al., 2012*) and a BLAST analysis to identify homologous sequences. A cluster analysis was done using inducible *HSP70* sequences described for other birds to identify conserved zones for primer design. Designed primers were 5′-TGACAAGTGCCGGGAGG-3′ (forward) and 5′-GGAGAAACTCTGCAACCCG-3′ (reverse). A 124-bp PCR product from blood cDNA of *P. rubricauda* was amplified, purified and sequenced in Macrogen Inc. (Korea). The sequence obtained showed a 99% similarity with *P. lepturus HSP70*. The same was done in order to design β-actin primers for *P. rubricauda*. Designed primers for β-actin were 5′-ATGGACTCTGGTGATGGTGTT-3′ (forward) and 5′-CTGTAGCCTCTCTCT GTCAGG-3′ (reverse). The β- actin gene was used as endogenous control for the qPCR analysis as the CT variation between samples was lower than 2 CT.

Template cDNA was used in a quantitative real-time PCR (RT-qPCR) using commercial fluorescent dye Takyon Low ROX SYBR 2x (Nalgene®, Rochester, NY, USA) to quantify transcriptional levels of *HSP70*. Reactions were run in triplicate in a Real-Time PCR System Agilent Technologies (Stratagene MX3000P). Reactions contained two μL of cDNA, and 18 μL of the master mix, which contained 10 μM of each primer, Takyon and Milli-Q water in a final volume of 20 μL. The initial denaturing time was 3 min at 95 °C, followed by 40 PCR cycles of 95 °C for 15 s and 60 °C for 30 s, followed by 95 °C and 55 °C and 95 °C for 15 s. After the PCR cycles, the purity of the PCR product was checked by the analysis of its melting curve; the thermal profile for melting curve analysis consisted of denaturation for 15 s at 95 °C, lowered to 55 °C for 15 s and then increased to 95 °C for 15 s with continuous fluorescence readings. During RT-qPCR, the efficiency of *HSP70* gene amplification was approximately equal to that of the housekeeping gene *β-actin* (as it was determined by slope calculation). Relative quantification of *HSP70* was determined by the comparative CT method (ΔΔCT method; *Livak & Schmittgen, 2001*).

## Body condition

Body condition was calculated with a Multiple Linear Regression of body mass as dependent variable with bill length, skull length, wing length and tarsus length as predictors following *Plischke et al. (2010)*. Based on backward elimination procedure, bill and skull length were discarded from the saturated model. An expected body mass was then calculated for each combination of the size factors as follows: Expected Body Mass = (1.724 × wing length) + (7.357 × tarsus length) − 26.038. Finally, body condition was calculated for each bird as a ratio between observed body mass and expected body mass.

## Molecular sexing

The red-tailed tropicbird is a monomorphic species that may show cryptic or subtle sexual dimorphism to the human eye (*Ismar et al., 2011*). A molecular sexing method using the DNA primers 2550F and 2718R (*Fridolfsson & Ellegren, 1999*) was performed to reliable determine the sex of the individuals. This sexing protocol has been used in previous studies in this species (*Ismar et al., 2011*; *Dehnhard & Hennicke, 2013*). Total genomic DNA was extracted from blood samples collected in FTA cards (Whatman paper, GE Healthcare Life Sciences) using the DNeasy Blood and Tissue Kit (Qiagen, Hilden, Germany). Reactions of 10 μL total volume consisted of ~100 ng of DNA, 1× PCR buffer (160 mM $(NH_4)_2SO_4$, 670 mM Tris–HCl, 0.1% stabilizer), 2 mM MgCl2, 0.6 μM of each primer, 0.25 mM of each dNTPs, 1 U of Taq polymerase, and 0.4 mg mL$^{-1}$ of Bovine Serum Albumin (BSA). PCR cycles were performed on an Agilent SureCycler 8800 Thermal Cycler, as follows: 94 °C for 2 min, followed by 35 cycles of 94 °C for 30 s, 50 °C for 45 s, 72 °C for 45 s, and a final extension at 72 °C for 10 min. Individuals were identified as female if visualized PCR products in agarose electrophoresis showed both the 450 bp CHD1W band and the 600 bp CHD1Z band, and as male if only the 600 bp CHD1Z band was present.

## Statistical analyses

All analyses were performed in R software (*R Core Team, 2017*), and figures were made using the package *ggplot2* (*Wickham, 2009*). We tested for the effect of sex (male or female) of the birds in the response variables. Because no significant effects were detected, this factor was excluded from further analyses. The three adults rearing chicks were excluded from statistical analyses because physiological responses can vary widely in relation to breeding stage in birds (*Hõrak et al., 1998*; *Wojczulanis-Jakubas et al., 2015*; among others). The birds that had neither eggs nor chicks were pooled with adults having eggs, after determination of no effects on the response variables. However, for modeling, an offset binary covariate was included to indicate whether the birds had an egg or not.

Two types of analyses were conducted. Firstly, we focused on determining the stress responses of the breeding birds. These analyses were conducted separately for the two islands because the explanatory variables were different between islands: (a) nests type (exposed vs. protected) for Rapa Nui and (b) louse flies' parasitic load (number of louse flies per bird) for Salas & Gómez. For both islands, the response variables were (a) the

five leukocyte types counted in blood smears, (b) the proportion of H and L leukocytes, (c) mRNA *HSP70* relative transcriptional level and (d) the body condition. To analyze the relationship between the explanatory variables for each island and the count of leukocyte cells, we used generalized multivariate models (GLMs). The models were constructed using the *manyglm* function available in the package *mvabund* (*Wang et al., 2012*), fitting a negative binomial distribution selected after model validation. Model significance was calculated using a Wald statistic test applying the *anova.manyglm* function with post-hoc univariate tests used to determine the responses of each leukocyte cell. Because the counts of leukocyte cells were obtained from the same sample of blood obtained from each bird, the independent assumption for cell counts is not satisfied. Thus, *p*-values were corrected by shrinking the sample correlation matrix towards identity (*Wang et al., 2012*) and using a Montecarlo method of resampling with 999 bootstrap iterations. The proportion of H and L leukocytes was modeled using a univariate logistic regression analysis using a GLM, assuming a binomial probability distribution. Multivariate Analysis of Variance (MANOVA) was performed to test for the response of both the mRNA *HSP70* relative transcriptional levels and body condition, between islands and within each colony. The best models for every response variable were selected in terms of the maximum likelihood criteria (*Crawley, 2007*). Model validations were done examining the normality of the residuals (Shapiro–Wilk normality test) generated by each model.

The second type of analysis consisted in determining the effect of the body condition of the birds at the level of the five leukocyte cells. A multivariate GLM was applied as described above using the *manyglm* function but using the body condition (expressed as natural logarithm) as the regressor for leukocyte cell counts, and island as a categorical factor. The model significance and validation were examined in the same way as described above.

## RESULTS

### Stress indicators and its relationship with nest type in the Rapa Nui colony

No significant differences were found in the count of any leukocyte type when comparing between protected and exposed nests (Deviance test = 1.779; $P = 0.830$). There were not significant differences between the two nest types in the H/L ratio (Residual Deviance = 19.889; $P = 0.330$), mRNA *HSP70* relative transcriptional level (MANOVA, $F = 0.088$, $P = 0.771$), nor body condition (MANOVA, $F = 3.495$; $P = 0.091$). Neither, the H/L ratio nor mRNA *HSP70* relative transcriptional levels showed a relationship to body condition ($r = -0.382$, $P = 0.198$ and $r = -0.522$, $P = 0.099$, respectively).

Eosinophils were the most abundant leukocyte in Rapa Nui, with a total mean of 36.69%, followed by heterophils (31.83%) and lymphocytes (30.85%) (Table 1). Monocytes and basophils were the less abundant with a mean of 5.88% and 0.63% respectively (Table 1). When analyzing for nest types, eosinophils were also the most abundant for both categories, with a mean of 40.71% in protected nests and a mean of 41.31% in exposed nests (Table 2).

**Table 1 Leukocyte analysis of the red-tailed tropicbird, *Phaethon rubricauda* from Rapa Nui and Salas y Gómez islands.** Mean estimated values, standard deviation (SD), range of leukocyte counts, heterophil/lymphocyte ratio (H/L ratio) and body mass (g) of red-tailed tropicbird adults assessed between August 2016 and June 2017 in Rapa Nui and Salas & Gómez islands.

| Leukocytes | Rapa Nui | | Salas & Gómez | |
|---|---|---|---|---|
| | *n* = 24[*] | | *n* = 22[*] | |
| | Mean ± SD | Range | Mean ± SD | Range |
| Heterophils % | 31.83 ± 14.63 | 4.00–59.00 | 33.73 ± 11.96 | 12.00–57.50 |
| Lymphocytes % | 30.85 ± 9.95 | 17.00–53.50 | 27.73 ± 8.44 | 15.50–48.00 |
| Eosinophils % | 36.69 ± 13.74 | 15.50–66.00 | 36.77 ± 11.41 | 11.00–55.50 |
| Monocytes % | 5.88 ± 2.34 | 1.50–10.50 | 4.36 ± 2.93 | 0.00–9.50 |
| Basophils % | 0.63 ± 0.90 | 0.00–3.50 | 1.77 ± 2.91 | 0.00–10.50 |
| H/L ratio | 1.18 ± 0.73 | 0.15–2.77 | 1.41 ± 0.77 | 0.29–3.43 |
| Heterophils/10,000 erythrocytes | 6.54 ± 5.04 | 0.48–17.82 | 11.79 ± 7.22 | 0.28–23.40 |
| Lymphocytes/10,000 erythrocytes | 6. 20 ± 3.66 | 0.71–14.14 | 7.56 ± 5.22 | 0.96–25.67 |
| Eosinophils/10,000 erythrocytes | 7.57 ± 6.10 | 1.66–27.06 | 10.27 ± 6.45 | 0.76–23.83 |
| Monocytes/10,000 erythrocytes | 1.14 ± 0.78 | 0.15–3.68 | 1.41 ± 1.29 | 0.00–4.12 |
| Basophils/10,000 erythrocytes | 0.10 ± 0.16 | 0.00–0.65 | 0.66 ± 1.27 | 0.00–4.57 |
| Total leukocytes/10,000 erythrocytes | 20.40 ± 11.36 | 4.33–52.52 | 29.26 ± 16.33 | 2.00–58.23 |
| Body mass (g) | 827.8 ± 55.7 | 740.0–960.0 | 828.9 ± 84.2 | 700.0–990.0 |

Note:
[*] Sample size after exclusion of samples due to deficient conditions of smears and of adults rearing chicks (two from Rapa Nui and three from Salas & Gómez). See Methods for details.

## Stress indicators and its relationship with louse flies' parasite load in the Salas & Gómez colony

All flies captured in this study were identified as *Olfersia* sp. (Diptera: Hippoboscidae) (following *Bequaert, 1941*; *Maa, 1966*; *Santos, López & Miller, 2014*). The median of louse flies' parasitic load was 4 (Range 0–26), with a prevalence of 91% of parasitized birds. Of 22 red-tailed tropicbird adults assessed in Salas & Gómez, two birds (9%) had no flies at the moment of assessment, seven (32%) had between 1 and 5 flies, 10 (45%) had between 6 and 11 flies, and only three birds (14%) had more than 22 flies. In most cases, the 5 min' limit was sufficient to capture almost all louse flies inhabiting the birds. In two cases the 5 min' limit was not sufficient to capture all the louse flies over the birds. These two birds had 22 and 26 flies each, and at least 10 flies were not captured when the time limit was reached. These extra counts were not added in order to maintain a standard methodology.

No significant differences were found in the count of any leukocyte type in relation to louse flies' parasitic load (Deviance test = 2.853; $P$ = 0.588). The louse flies' parasitic load did not had a significant effect in the H/L ratio (Residual deviance = 20.562; $P$ = 0.521), the mRNA *HSP70* relative transcriptional level (MANOVA, $F$ = 0.338; $P$ = 0.569) nor body condition (MANOVA, $F$ = 1.212; $P$ = 0.286). Neither, the H/L ratio nor mRNA *HSP70* relative transcriptional levels showed a relationship to body condition ($r$ = −0.2217, $P$ = 0.2978; and $r$ = 0.2946, $P$ = 0.1832, respectively).

**Table 2 Leukocyte analysis of the red-tailed tropicbird, *Phaethon rubricauda* at two nest categories in Rapa Nui.** Mean estimated values, standard deviation (SD), range of leukocyte counts, heterophil/lymphocyte ratio (H/L ratio), and body mass (g) of red-tailed tropicbird adults assessed on June 2017 in Rapa Nui at protected and exposed nests.

| Leukocytes | Protected | | Exposed | |
|---|---|---|---|---|
| | $n = 7$* | | $n = 8$* | |
| | Mean ± SD | Range | Mean ± SD | Range |
| Heterophils % | 26.29 ± 15.17 | 4.00–45.50 | 31.44 ± 14.92 | 9.00–49.50 |
| Lymphocytes % | 32.57 ± 7.61 | 24.00–44.00 | 26.19 ± 11.31 | 17.00–53.00 |
| Eosinophils % | 40.71 ± 14.36 | 28.00–66.00 | 41.31 ± 11.87 | 27.50–61.00 |
| Monocytes % | 6.86 ± 2.08 | 4.50–10.00 | 6.25 ± 2.78 | 1.50–10.50 |
| Basophils % | 0.43 ± 0.61 | 0.00–1.50 | 1.06 ± 1.32 | 0.00–3.50 |
| H/L ratio | 0.90 ± 0.67 | 0.15–1.85 | 1.43 ± 0.88 | 0.17–2.77 |
| Heterophils/10,000 erythrocytes | 4.23 ± 3.02 | 0.48–8.86 | 5.58 ± 4.86 | 1.72–16.92 |
| Lymphocytes/10,000 erythrocytes | 5.04 ± 3.26 | 1.95–12.01 | 4.81 ± 2.84 | 0.71–10.14 |
| Eosinophils/10,000 erythrocytes | 7.01 ± 6.39 | 1.69–20.82 | 7.69 ± 4.88 | 1.66–17.59 |
| Monocytes/10,000 erythrocytes | 1.01 ± 0.50 | 0.43–1.88 | 1.04 ± 0.51 | 0.15–1.73 |
| Basophils/10,000 erythrocytes | 0.04 ± 0.05 | 0.00–0.10 | 0.19 ± 0.24 | 0.00–0.65 |
| Total leukocytes/10,000 erythrocytes | 16.31 ± 10.52 | 4.71–37.51 | 18.27 ± 9.36 | 4.33–34.22 |
| Body mass (g) | 831.4 ± 45.9 | 770.0–890.0 | 805.6 ± 55.9 | 750.0–920.0 |

Note:
* Sample size after exclusion of samples due to deficient conditions of smears (one) and of an adult rearing a chick (one). See Methods for details.

Eosinophils were the most abundant leukocytes in birds from Salas & Gómez, with a total mean of 36.77%, followed by heterophils (33.73%) and lymphocytes (27.73%) (Table 1). Monocytes and basophils were the least abundant with a mean of 4.36% and 1.77% each (Table 1).

## Comparison between islands

Comparison of the H/L ratio between islands showed no significant differences (Residual deviance = 53.347; $P = 0.0644$). Similarly, the mRNA *HSP70* relative transcriptional levels and body condition showed no significant differences when comparing between islands (MANOVA, $F = 0.080$, $P = 0.778$ and $F = 0.0034$, $P = 0.954$, respectively; Fig. 1).

The regression analysis using *mvabund* revealed a significant effect between islands in the count of leukocytes regressed to body condition (Wald test = 4.810, $P = 0.001$, Table 3). Post-hoc univariate test showed an opposite trend between islands in eosinophil, lymphocyte, and monocyte counts related to body condition ($P = 0.014$; $P = 0.027$, and $P = 0.027$, respectively, Table 3). These three leukocytes showed a positive trend when related to body condition for Rapa Nui birds, while a negative trend was detected for Salas & Gómez (Fig. 2). In the case of heterophils and basophils no significant effects were detected ($P = 0.479$, and $P = 0.207$, respectively, Table 3; Fig. 2).

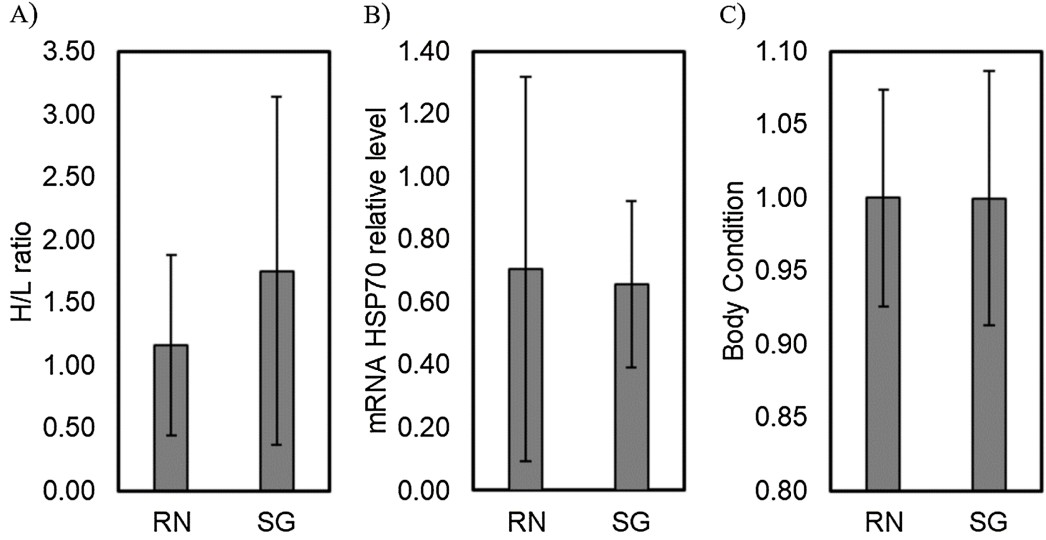

**Figure 1 Comparison of physiological stress traits between colonies.** Levels of physiological stress traits, (A) H/L ratio, (B) mRNA HSP70 relative transcriptional level and (C) body condition of red-tailed tropicbirds assessed in Rapa Nui (RN) and Salas & Gómez (SG) islands. Bars represent means ± standard deviation.

## DISCUSSION

There are several physiological studies in birds focused on passerine nestling immunological response against a wide variety of stressors, including nutritional constraints and parasite infestation (*Saino, Calza & Møller, 1998*; *Lobato et al., 2005*; *Masello et al., 2009*; *Martínez-de la Puente et al., 2011*). However, there is a lack of studies to understand the adult's physiological responses of some well-documented threats to seabirds (e.g., predation by introduced predators and other anthropogenic perturbations). This is the first physiological study performed to evaluate the effects of both the risk imposed by introduced predators and parasites on selected physiological traits in adults of the red-tailed tropicbird *Phaethon rubricauda* from the eastern Pacific Ocean.

It was expected that the level of threats imposed by introduced predators in the Rapa Nui colony (*Flores et al., 2017*; *Luna et al., 2018*), and the level of infestation by louse flies in Salas & Gómez would be reflected in responses in the H/L ratio (*Saino, Calza & Møller, 1998*; *Caetano et al., 2014*), in the mRNA *HSP70* level (*Morales et al., 2004*; *Thomson et al., 2010*) and body condition (*Scheuerlein, Van't Hof & Gwinner, 2001*) in red-tailed tropicbirds. However, no differences were found in any of the stress level indicators assessed when comparing between protected and exposed nests in Rapa Nui, or among birds with different louse flies' parasitic loads in Salas & Gómez. Similarly, we found a lack of significant difference in the H/L ratio, the mRNA *HSP70* level and body condition when comparing between islands. Nevertheless, a significant differentiation between islands was detected in the eosinophil, lymphocyte and monocyte counts when related to body condition. Because these measures were done under natural scenarios, it was not possible to evaluate colonies free of introduced predators or parasites.

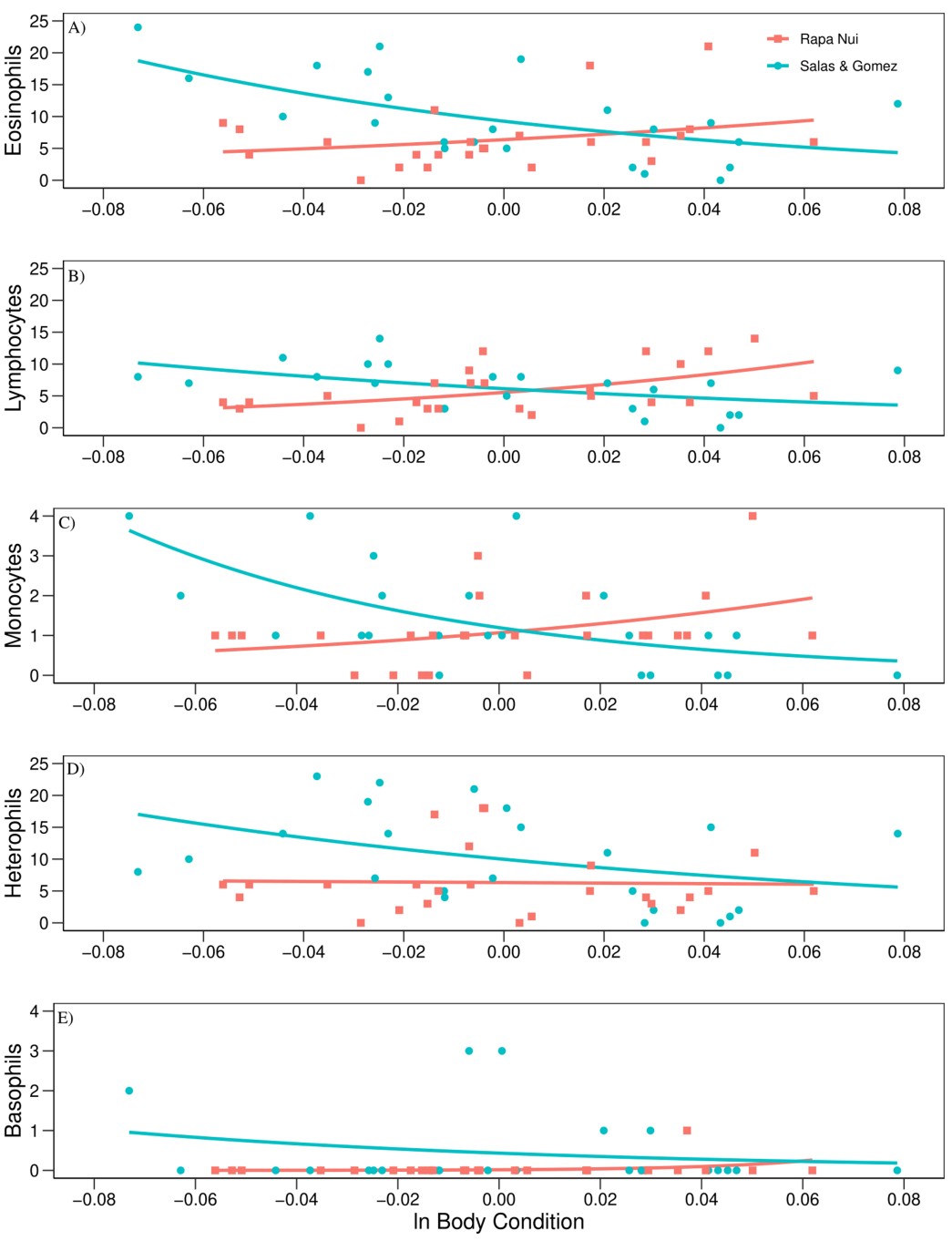

**Figure 2 Count of leukocytes regressed to body condition.** Relationship among (A) eosinophils, (B) lymphocytes, (C) monocytes, (D) heterophils, (E) basophils, and ln body condition of adult red-tailed tropicbirds assessed in Rapa Nui and Salas & Gómez islands.

Heterophils and lymphocytes are the most abundant leukocyte types among bird's haematological profiles, composing about 80% of total leucocyte counts (*Davis, Maney & Maerz, 2008*; *Clark, Boardman & Raidal, 2009*). It is noteworthy that eosinophils were importantly elevated in this study, being as abundant as heterophils and lymphocytes,

**Table 3 Generalized linear models for leukocytes counts.** Multivariate and univariate generalized linear models for leukocytes counts of red-tailed tropicbird adults assessed in Rapa Nui and Salas & Gómez islands. The cell counts were regressed against the birds' body condition for each island (see Fig. 2). Significant values are shown in bold. See "Materials and Methods" for details about modeling.

**Multivariate test**

|  | | Res.Df | Df. Diff | Wald test | P |
|---|---|---|---|---|---|
| | (A) Body Condition (log) | 46 | 1 | 2.783 | 0.160 |
| | (B) Islands | 45 | 1 | 2.863 | 0.103 |
| | A × B | 44 | 1 | 4.810 | **0.001** |

**Univariate tests**

| | Eosinophils | | Lymphocytes | | Monocytes | | Heterophils | | Basophils | |
|---|---|---|---|---|---|---|---|---|---|---|
| | Wald test | P | Wald test | P | Wald test | P | Wald test | P | Wald test | P |
| (A) Body Condition (log) | 1.475 | 0.310 | 0.859 | 0.459 | 1.754 | 0.239 | 2.297 | 0.078 | 1.065 | 0.459 |
| (B) Islands | 1.195 | 0.517 | 0.568 | 0.688 | 0.700 | 0.688 | 1.350 | 0.493 | 2.296 | 0.058 |
| A × B | 3.190 | **0.014** | 2.704 | **0.027** | 2.814 | **0.027** | 0.650 | 0.479 | 1.257 | 0.207 |

reaching abundances of 35% on average, in contrast to the average <2% previously reported for red-tailed tropicbirds (*Dehnhard, Quillfeldt & Hennicke, 2011*; *Dehnhard & Hennicke, 2013*). This unusual abundance was observed in both colonies assessed in the present study, suggesting that eosinophils are elevated in red-tailed tropicbirds inhabiting this ecoregion. While decreased numbers of eosinophils are related to high stress conditions, high numbers have been related to an immune reaction against intestinal parasite loads (*Hawkey et al., 1983*; *Maxwell, 1993*; *Davis, Maney & Maerz, 2008*; *Dehnhard & Hennicke, 2013*). It is thus possible that unnoticed intestinal parasites are present in both colonies causing this high abundance of eosinophils. However, further studies are necessary to elucidate the exact origin of this observation.

It has been reported that blood parameters commonly differ in relation to incubation stage in birds (*Hõrak et al., 1998*; *Jakubas, Wojczulanis-Jakubas & Kreft, 2008*; *Hrabcakova et al., 2014*; *Wojczulanis-Jakubas et al., 2014*; *Wojczulanis-Jakubas et al., 2015*). However, we were unable determine the laying date of the breeding birds to discriminated between early and late incubation stage to take this into account in our analyses. Fieldwork was conducted in a very limited period within the breeding season and a long-term monitoring of the assessed colonies was not possible due to their isolation and restricted access. On the other hand, we were able to examine the potential effect of bird's sex in our analyses. We found that the sex of the birds does not affected any of the variables assessed. A similar result was reported for a red-tailed tropicbird colony at Christmas Island on the Indian Ocean, where neither leucocyte profiles nor body condition differed significantly between sexes (*Dehnhard & Hennicke, 2013*). This may suggest that red-tailed tropicbird males and females equally contributes to the energetic demands of the breeding period (*Dehnhard & Hennicke, 2013*; *Wojczulanis-Jakubas et al., 2015*).

The lack of differences in any of the physiological stress indicators assessed on adult birds between exposed and protected nests within the Rapa Nui colony may be explained by the absence of differences in the frequency of introduced predators visits between nests types, as found in the experiment with simulated unattended eggs in this colony (*Luna et al., 2018*). It is worth noting that in the experiment reported in *Luna et al. (2018)* simulated eggs were placed in empty red-tailed tropicbird nests when no birds were present in the colony. In the present study we further evaluated the potential differences between the two nest categories because the presence and behavior of introduced predators may differ widely between an active vs. an un-active colony. However, both studies seem to suggest that in this colony terrestrial predators visit all nests equally, and aerial predators may not increase predation risk in exposed nests. Even though in the present study each of the active nests were sampled during the timeframe of the fieldwork, further research including a bigger sample size will be necessary to corroborate these results.

The lack of correlation between different levels of louse flies' parasitic loads and the assessed stress traits on the Salas & Gómez colony may be explained by the behavior of this ectoparasite. During sampling, it was observed that louse flies tended to leave their hosts when these flew away for feeding; and reattached to them when they come back to the colony. This behavior of the flies may have affected the accuracy in assessing the real parasitic load, and thus affecting results on stress indicators levels. The method used to measure the parasitic load of louse flies in this study should thus be considered as the first evidence of the high prevalence of parasites in this colony, but may not reflect the accumulated parasitic load effect on stress over time.

A possible explanation for the absence of differences in stress indicators between the two assessed colonies in this study might be related to the presence of the native Great Frigatebird (*Fregatta minor*) in Salas & Gómez. This species is known to kleptoparasitize on several seabird species, including the red-tailed tropicbird, and also is reported to depredate on small seabird chicks (*Schreiber & Ashmole, 1970*; *Megyesi & Griffin, 1996*). Red-tailed tropicbirds in Salas & Gómez nests under rocky protection, and no exposed nests were observed in this colony. This rise the question if this nesting behavior is because of soil's conditions, or if birds prefer protected nests to avoid avian predation by great frigatebirds. Predation of chicks by great frigatebirds has never been reported on any species in Salas & Gómez, likely due to a lack of observations because of its remote location. Despite this, the presence of this species on a small island like Salas & Gómez, surrounded by oligotrophic oceanic conditions (*Morel, Claustre & Gentili, 2010*), may raise the risk of frigatebirds attempting to depredate on red-tailed tropicbird chicks, with probably similar effects on physiological stress indicators, as introduced predators in Rapa Nui.

The only significant difference found in this study was in the eosinophil, lymphocyte, and monocyte counts when related to body condition between islands. Although body condition per se was not a significant factor within islands, the interaction between the two

factors (body condition and island) showed a significant effect in the trend of these three leukocytes. For the Rapa Nui colony, the trend was an increase of these three leukocytes with a better body condition. The opposite trend was found for the colony at Salas & Gómez, where increased levels of these three leukocytes was related to a poorer body condition. Eosinophils, lymphocytes and monocytes are all involved in the immune response against infections. It may be possible that birds from Salas & Gómez in a poorer body condition where more susceptible to an infestation by louse flies, together with unnoticed intestinal parasites. Furthermore, Haemosporidian blood parasites are usually transmitted by louse flies and cause infection (*Rahola, Goodman & Robert, 2011*). These blood parasites were not detected in this study despite of an exhaustive visual examination of blood smears. However, we cannot discard an undetected presence of these parasites in the assessed birds. Genomic techniques may be necessary to detect Haemosporidian parasites from bird's blood samples (*Cassin-Sackett, 2020*). It is also possible that an infection by parasites is the cause of the poorer condition of some birds at Salas & Gómez and not that birds in a poorer body condition are more affected by an infection. On the other hand, our results may suggest that birds at Rapa Nui in a better body condition showed a higher immune activity compared to birds in a poorer condition. As stated before, eosinophils were elevated in both colonies, suggesting an infection by unnoticed intestinal parasites in this ecoregion. It may be the case that an additional infection by louse flies—and possible by Haemosporidian blood parasites- at Salas & Gómez is the cause for the significant difference in the trend of eosinophils, lymphocytes, and monocytes between the islands. However, it should be acknowledged that the colonies were assessed under a natural scenario and several other factors may have influenced the difference of these three leukocytes when related to body condition between the two colonies.

## CONCLUSIONS

We found a lack of significant differentiation in all the stress level indicators assessed within islands (protected vs. exposed nests in Rapa Nui and different louse flies' parasitic loads in Salas & Gómez) and between islands. However, this study revealed a significant opposite trend between islands in the eosinophil, lymphocyte, and monocyte counts when related to body condition. The presence of louse flies in Salas & Gómez vs. the absence of this parasite at Rapa Nui may be the cause for the significant difference in the trend of these three leukocytes between the islands. Studies of both, intestinal and blood parasites presence in these two colonies, and on the potential effects of other unassessed factors could contribute to better understand the difference found in these three leukocyte counts when related to body condition between colonies and the unusual high eosinophil count for birds in this ecoregion.

Even though no significant effect of predator's presence was observed on the physiological condition of adult red-tailed tropicbirds (sub-lethal interaction), attention should be paid on the direct effect of predators (prey mortality) over the Rapa Nui's colony, especially on the vulnerability of unattended chicks and fledglings to predation during the breeding seasons.

# ACKNOWLEDGEMENTS

We are grateful to the Chilean Navy for transport to Salas & Gómez Island. We also express our gratitude to the indigenous community of Ma'u Henua and to the National Forest Corporation-Rapa Nui (CONAF-Rapa Nui) for allowing access to the colony at Rano Raraku at the Rapa Nui National Park. We would like to acknowledge Pedro Lazo and Graciela Campbell (CONAF-Rapa Nui rangers) for their help during fieldwork. Miriam Lerma and Juan Serratosa (PhD students) collected the samples from Salas & Gómez in one of the three sampling trips to the island. Finally, we thank to the Agricultural and Livestock Service (SAG), Chile, for providing the necessary permits for bird assessments.

### Funding

Funding for this project was provided by a postdoctoral research grant awarded to Andrea I. Varela (CONICYT-FONDECYT N 3160324), by a MSc scholarship awarded to Nicolas Luna (CONICYT N 22161894), and by the Millenium Nucleus for Ecology and Sustainable Management of Oceanic Islands (ESMOI), a Scientific Initiative supported by the Ministry of Economy, Development and Tourism (Chile). The funders had no role in study design, data collection and analysis, decision to publish, or preparation of the manuscript.

### Grant Disclosures

The following grant information was disclosed by the authors:
CONICYT-FONDECYT N: 3160324.
CONICYT N: 22161894.
Millenium Nucleus for Ecology and Sustainable Management of Oceanic Islands (ESMOI).
Ministry of Economy, Development and Tourism (Chile).

### Competing Interests

The authors declare that they have no competing interests.

### Author Contributions

- Nicolas Luna performed the experiments, analyzed the data, prepared figures and/or tables, authored or reviewed drafts of the paper, and approved the final draft.
- Andrea I. Varela conceived and designed the experiments, performed the experiments, authored or reviewed drafts of the paper, and approved the final draft.
- Guillermo Luna-Jorquera conceived and designed the experiments, analyzed the data, authored or reviewed drafts of the paper, and approved the final draft.
- Katherina Brokordt conceived and designed the experiments, authored or reviewed drafts of the paper, and approved the final draft.

## Animal Ethics

The following information was supplied relating to ethical approvals (i.e., approving body and any reference numbers):

The Agricultural and Livestock Service (SAG), Chile, approved the manipulation of the birds (Resolucion exenta N°:9894/2015; Resolucion exenta N°:5343/2016).

## Ethics

The following information was supplied relating to ethical approvals (i.e., approving body and any reference numbers):

Ethics committee of the Universidad Católica del Norte, Coquimbo, Chile granted Ethical approval to carry out this study (Resolución F.M. N°12).

## Field Study Permissions

The following information was supplied relating to field study approvals (i.e., approving body and any reference numbers):

The indigenous community of Ma'u Henua and the National Forest Corporation-Rapa Nui (CONAF-Rapa Nui) allowed access to the colony at the Rapa Nui National Park.

## Data Availability

All the data collected in this study is available in the Supplemental Files.

## Supplemental Information

Supplemental information for this article can be found online at http://dx.doi.org/10.7717/peerj.9088#supplemental-information.

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
