# Peer review of "Effect of predation risk and ectoparasitic louse flies on physiological stress condition of the red-tailed tropicbird (Phaethon rubricauda) from Rapa Nui and Salas & Gómez islands"

_PeerJ, doi:10.7717/peerj.9088_

## Round 0.1 · original submission · Major Revisions

· Academic Editor

Major Revisions

Dear Dr. Luna and colleagues:

Thanks for submitting your manuscript to PeerJ. I have now received two independent reviews of your work, and as you will see, the reviewers raised some concerns about the research. Despite this, the reviewers are optimistic about your work and the potential impact it will have on research communities studying the biology and ecology of the red-tailed tropicbird. Thus, I encourage you to revise your manuscript, accordingly, taking into account all of the concerns raised by both reviewers.

In your revision, please address the specific concerns regarding the experimental design and data analyses, including lack of controls. Also, it appears that the sex of the study animals is not provided. Sex alone could have important consequences on all of the parameters that you have measured. If you restricted the study to females, then no new stats will have to be run. However, if you have used both sexes then you really need to test for sex-based effects in your model.

I look forward to seeing your revision, and thanks again for submitting your work to PeerJ.

Good luck with your revision,

-joe

·

Basic reporting

The manuscript is well written, concisely and clearly presented. The introduction is clear and cover the topic with enough detail. I have however a few suggestions for improvement.
1.1. Some references are poor/inappropriate. Creswell 2008 in line 75 deals with predation, while a proper reference dealing with parasitism is needed instead here. Aramburú 2012 do not seem the best reference in several parts where it is cited, and a more general/relevant reference should be cited instead.
1.2. Too many tables and figures are presented, all but one figure depict non-significant results. Consider removing some of the figures.

Experimental design

The study system is worth of investigation, as well as the questions tackled. Understanding the effects of predation and parasitism on the ecology of this bird species is especially relevant given its limited distribution. I have however several concerns regarding the experimental design and data analyses.
2.1. Information on breeding stage of sampled birds is not known (apart from incubating / rearing chicks). Nevertheless, the incubation process is an energy demanding process that affects the physiology of the birds. Birds in the early incubation stage will be different in many aspects to birds in the late incubation stage. Thus, this has probably introduced noise in the data, making more difficult to find relationships between the physiological variables explored with predation risk and parasitism. I am aware that this info may not be available in this system, but this concern should be discussed in the manuscript. Moreover, given that samples were collected during a relatively extended period of time, spanning from June to November, consider including date in the models somehow, to take into account this seasonal / phenological variability. In addition, only three of the birds were rearing a chick; would results change if these three nests are excluded from the analyses? (The only two birds from Rapa Nui with louse flies could be also excluded as well). The same concern regarding incubation stage applies to bird sex, which is not known and is an important source of noise that may hinder detection of the expected relationships.
2.2. I have concerns that the design may be flawed if, as stated in the ms (lines 390-395) predation risk is the same between exposed and protected nests. Please explain much better what these two categories mean.
2.3. In addition, a few more details are needed in the methods: State for example whether the bird nests colonially, which is important as it may determine parasitism exchange among neighbour nests, while sharing predation risk as well.

Validity of the findings

3.1. I want to stress that it is difficult to find effects of parasitism and predation risk even in more controlled and standardized conditions, so it may not be surprising that no such effects have been detected in this study. First, it is a correlative study, with no experimental manipulations performed. Second, the distinction between nests exposed and protected from predators may not be clear (see my previous comment above).
3.2. Apparently, the only relevant finding of the study is that H/L ratio is lower in Rapa Nui island. All other results are nonsignificant, although some of them are reported as “not significant trends”. These terminology is justified when p values approach significance, but not in all other cases; please rewrite these sections.
3.3. Info on percentages of leukocytes is not so relevant as to start each section of the results with it.
3.4. Interpretation of the difference in H/L ratio between islands is attributed to louse flies. However, the two islands differ in many other aspects, as the authors already point out (frigatebird presence, nutrient abundance around the island,…). Thus, without having detected an effect of ectoparasites on H/L ratio, it is not justified to defend this as the cause of the difference between islands. Please reconsider this interpretation.
3.5. Authors suggest that blood parasites may be involved as well in the observed difference in H/L ratios, although they have no data on blood parasites. Nevertheless, during microscope slide examination to conduct leukocyte counts, authors have to know whether there are or there are not blood parasites in the sampled birds (at least the most common haemosporidia).

Additional comments

Minor comments:

L267: Info repeated three lines before, please rewrite/delete.
L338: Delete “at the moment of assessment”
L424: Could this huge difference in percentage of eosinophils be due to misidentification in one of the two studies?

Reviewer 2 ·

Basic reporting

Abstract:
Line 31: “and/or” would be better here.
Line 45: would replace “categories” and provide a more detailed description here. Either exposed on unexposed nests.
Line 48: Compare to what? Birds from Rapa Nui?
Line 57: raptors birds
Line 92: I would rather see the authors state that blood smears are more commonly used to investigate the immune status of an individual. Stress is most commonly investigated using circulating levels of glucocorticoids which is known to mediate the physiological response to challenging conditions. Smears are much more specific, in my opinion, to immune function. Immune function is one of the many factors affected by glucocorticoids. Deviations in measurements from blood smears do not have to be a direct response to an environmental stressor.
Line 173: Is the sex known for these birds? I am assuming that these are all females. Do you know the sample date relative to lay date? Females undergo wide fluctuations in blood parameters around the time of egg lay which could influence your measurements. Some authors that have looked at blood parameters around the laying period are Tony Williams, Jesse Krause, Michael Kern, P. Hõrak,,
and there is a nice review by Jeanne Fair. Also circulating glucocorticoids in females can be quiet variable around the transition from egg lay to incubation and feeding of young. Thus, changing circulating levels of glucocorticoids could be directly influencing the transcription of HSPs
Line 194: Is it known if HPSs are rapidly affected by handling? Meaning the delay in collecting the blood sample could affect the measurements. Perhaps it would be best to report your average time to blood sampling with standard deviation.
Line 327 & 354: If there is no relationship then do not state that there was a trend.
Line 408-421: It may be good to offer an alternative explanation. I agree that higher immune function induced by parasitic flies is likely a very plausible explanation. However, without a control it is hard to know which group is up and the other down. One could argue that stress is upregulated, just not in anything you measured, which results in immunosuppression of tropic birds on island inhabited by predators. Perhaps corticosterone levels are different between birds on the different islands which is directly affecting leucocytes. I would suggest adding this alternative explanation as well.

Experimental design

Luna et al examined red-tailed tropic birds to understand the effects of predation pressure and parasite load on various physiological markers that included heat shock proteins (HSPs), heterophile to lymphocyte ratios on two separate islands. The authors did not detect any significant correlations between H/L ratios or HSP and parasite load or body condition. MANOVA analysis were also not significant in determining the effects of the various continuous variables on nest site. There was a significant difference between H/L ratios between the two islands with high parasite load results in relatively higher H/L ratio.

Validity of the findings

Overall, this study is straight forward, from the data collection, to analysis, and data interpretation. One critical detail that is missing is the sex for the individuals that were sampled. It would be expected that sex effects especially when considering the sample date with egg lay date could influence these parameters. It would be greatly appreciated if the authors could clarify these details. It is unfortunate that there is not a control island in this study to truly understand the directionality of the significant relationships. The authors do acknowledge this short coming and to a certain extent is overcome by the physiological questions that are asked for each respective island before making the direct comparisons between the islands. Making comparisons between two treatments, essentially, makes it difficult to interpret which island truly has a higher or lower physiological parameter since it is not known what the unmanipulated phenotype would be.
I am always cautious in using the work stress in my writing because of its complexity. I do not disagree that the authors have investigated things that are directly affected by stress. I wonder if there is a better way of phrasing this study?

---

## Round 0.2 · accepted · Accept

· Academic Editor

Accept

Dear Dr. Luna and colleagues:

Thanks for re-submitting your revised manuscript to PeerJ, and for addressing the concerns raised by the reviewers. I now believe that your manuscript is suitable for publication. Congratulations! I look forward to seeing this work in print, and I anticipate it being an important resource for research communities studying the biology and ecology of the red-tailed tropicbird..

Thanks again for choosing PeerJ to publish such important work.

-joe

Reviewer 2 ·

Basic reporting

no comment

Experimental design

no comment

Validity of the findings

no comment

Additional comments

The authors have addressed all of my previous concerns.